# Enabling Arbitrary Translation Objectives with Adaptive Tree Search

**Wang Ling**[*][†]
Talka, Inc.
`lingwang@talka.ai`

**Wojciech Stokowiec**[*]**, Domenic Donato, Laurent Sartran,
Lei Yu & Chris Dyer**
DeepMind, Ltd.
`{wstokowiec,domenicd,lsartran,leiyu,cdyer}`
`@deepmind.com`

**Austin Matthews**
Amazon.com
`tinaus@amazon.com`

## Abstract

We introduce an adaptive tree search algorithm, which is a deterministic variant of Monte Carlo tree search, that can find high-scoring outputs under translation models that make no assumptions about the form or structure of the search objective. This algorithm enables the exploration of new kinds of models that are unencumbered by constraints imposed to make decoding tractable, such as autoregressivity or conditional independence assumptions. When applied to autoregressive models, our algorithm has different biases than beam search has, which enables a new analysis of the role of decoding bias in autoregressive models. Empirically, we show that our adaptive tree search algorithm finds outputs with substantially better model scores compared to beam search in autoregressive models, and compared to reranking techniques in models whose scores do not decompose additively with respect to the words in the output. We also characterise the correlation of several translation model objectives with respect to BLEU. We find that while some standard models are poorly calibrated and benefit from the beam search bias, other often more robust models (autoregressive models tuned to maximize expected automatic metric scores, the noisy channel model and a newly proposed objective) benefit from increasing amounts of search using our proposed decoder, whereas the beam search bias limits the improvements obtained from such objectives. Thus, we argue that as models improve, the improvements may be masked by over-reliance on beam search or reranking based methods.

## 1 Introduction

Conditional text generation tasks, such as machine translation, consist of two parts: a model that assigns scores to candidate outputs, and a search component that interacts with the model in order to find an output that maximizes the score assigned by the model. This search problem is a hard combinatorial optimization problem, and as a result, constraints are frequently imposed on the structure of the model to make solving or approximating the search problem easier. In neural machine translation, an autoregressive factorization of the output probability distribution is widely used (Kalchbrenner & Blunsom, 2013; Sutskever et al., 2014; Vaswani et al., 2017), and a variety of conditional independence assumptions are made in other model classes from statistical translation models (Brown et al., 1993; Koehn et al., 2003) to non-autoregressive neural models (Lee et al., 2018). Although these assumptions enable fast and accurate approximations to the search problem with simple and efficient algorithms (e.g., beam search), which can be crucial for efficient production applications, they limit the form of the models and thereby restricting the kinds of architectures that can be used to address observed model failures.

---

[*]Authors contributed equally.    [†] Work carried out while Wang Ling was at DeepMind.

Despite the algorithmic benefits that beam search provides, we argue that it is a poor foundation for long-term scientific progress toward an accurate and reliable translation model whose scores adequately predict translation quality. First, it can only be applied to autoregressive models, which have well-known length calibration problems due to a tendency to drop or "hallucinate" content, and this tendency has been remarkably resistant to remedy across a variety of different autoregressive architectures (Koehn & Knowles, 2017; Lin et al., 2020). The existing heuristic solutions—e.g., non-local corrections to the search objective at decoding time, or global statistics about the population of per-word probabilities (Wu et al., 2016; Meister et al., 2020)—point to non-autoregressive components in the search objective as necessary parts of the solution.[1] We therefore would like to directly work with model classes that contain these solutions, rather than being dependent on limited heuristics that are imposed after the fact. Second, beam search is strongly biased towards translations that yield high initial scores due to the heuristic used to score partial translations (Stahlberg & Byrne, 2019; Meister et al., 2020). While this feature provides short term translation quality gains as it addresses many of the calibration issues in autoregressive models, it is undesirable in the longer term as it masks many modeling issues that need addressing. More importantly, as scientific progress drives better correlations between model scores and translation quality, search errors caused by this bias will inevitably have a negative impact on both model scores and translation quality.

To address these shortcomings, we introduce the **beam adaptive tree search** (BATS) algorithm, which is based on Monte Carlo tree search (Coulom, 2006; Browne et al., 2012). Like MCTS, BATS estimates the value of internal nodes (i.e., partial translations) in the search tree with estimates from an expected-outcome model based on playouts from an auxiliary model, and these estimates are refined as the search progresses. Because BATS is guided from the start by the true objective—whether autoregressive or not, and due to the refinement of initial score estimates, the BATS decoder exhibits fewer biases as the search budget increases than beam search or reranking algorithms.

Our experimental section aims to characterise the impact of decoding mechanisms on both non-autoregressive and autoregressive models. For autoregressive models, we show that the calibration issues in autoregressive models can be addressed by adding a non-autoregressive component that augments the autoregressive sequence score with the lowest scoring produced token in the autoregressive decomposition of the sequence, which we name **max rank**. We further show that existing non-autoregressive approaches, such as the "noisy channel" model (Yu et al., 2017; Yee et al., 2019; Ng et al., 2019; Yu et al., 2020b;a; Liu et al., 2021), which factorize the translation probability according to Bayes' rule and have been argued to be better calibrated, are (1) poorly optimized by standard reranking-based approaches, and (2) ultimately have similar calibration failures as neural autoregressive models have. Incorporating the max rank component to existing objectives benefits both beam search and BATS, but the latter yields both the best translation and model scores. Crucially, we show that our decoder can search substantially longer and achieve higher model scores before BLEU starts to deteriorate, which suggests a negative impact of the search bias in beam search. Finally, we show that once autoregressive models become robust enough to address the calibration issues, this bias has an equally negative impact on translation quality. By fine-tuning an autoregressive model to better correlate with BLEU using minimum risk training (Shen et al., 2016), we show that BATS can achieve higher translation quality and a better model score compared to beam search.

## 2 BACKGROUND

### 2.1 BEAM SEARCH

Translation tasks operate on the space of possible translations $\mathcal{Y} = \Sigma^* \circ \{\text{EOS}\}$ where $\Sigma$ is a finite vocabulary, and EOS is a symbol represents the end of a sequence. Elements in $\mathcal{Y}$ correspond to full sentences $\boldsymbol{y} = y_1, \ldots, y_n$ with $n-1$ tokens, where $y_n = \text{EOS}$ (end of sentence). Each translation $\boldsymbol{y}$ is conditioned on a source sentence $\boldsymbol{x}$ and is assigned a score $s(\boldsymbol{x}, \boldsymbol{y})$, which measures how well $\boldsymbol{x}$ translates to $\boldsymbol{y}$. Decoding algorithms aim to find the best hypothesis $\boldsymbol{y}^* \in \mathcal{Y}$ under the **search objective** $s(\boldsymbol{x}, \boldsymbol{y})$. While autoregressive models may assign scores to prefixes of translations, we

---

[1]In this paper, we will use the term **non-autoregressive** to refer to any model whose scores do not decompose additively with the words in the output sequence. These include models that make conditional independence assumptions and generate each word independent of the others (Lee et al., 2018), but also energy based models that require a complete translation hypothesis to compute a score, and models that make a Bayes' rule decomposition of the translation probability.

only assume that $s$ is well defined for full translations, $\boldsymbol{y} \in \mathcal{Y}$. In the particular case of machine translation under standard autoregressive models, the search objective is defined as the conditional log probability:

$$s_{\text{AR}}(\boldsymbol{x}, \boldsymbol{y}) = \log p(\boldsymbol{y} \mid \boldsymbol{x}) = \sum_{i=1}^{|\boldsymbol{y}|} \log p(y_i \mid y_1, \dots, y_{i-1}, \boldsymbol{x}).$$

As autoregressive models have probability emissions at the token level, the search problem can be cast into a shortest path problem for weighted graphs, where each node $p$ is identified by a sequence $\boldsymbol{y}^{(p)}$. Nodes define an additional space $\mathcal{Y}^*$ including $\mathcal{Y}$ as well as **partial translations** in $\Sigma^*$ that do not terminate with a EOS token. Each node contains $|\Sigma| + 1$ edges, each appending a different word $y_i \in \Sigma \cup \{\text{EOS}\}$ to $\boldsymbol{y}^{(p)}$ generating a new node $p \circ y_i$. Edge weights are given by the emission log probability $\log p(y_i \mid \boldsymbol{y}^{(p)}, \boldsymbol{x})$ according to the autoregressive model. Search starts from the root node $\varepsilon$ with an empty sequence and nodes that generate the EOS token have a single edged with weight 0 that lead to a single shared terminal node. One solution to this problem is A$^*$ search (Hart et al., 1968), which is a best-first algorithm that iteratively searches nodes $p$ with the highest value $\hat{v}(p) = c(p) + f(p)$, where $c(p)$ is the sum of weights of all edges from $\varepsilon$ to $p$, and $f(p)$ is a heuristic function that attempts to estimate the sum of the edges on the highest scoring path from $p$ to the terminal node.

Although theoretically appealing, good A$^*$ heuristics are difficult to obtain, and search can be extremely expensive with poor heuristics. To remedy this, **beam search** introduces approximations. Rather than a best-first traversal, beam search proceeds iteratively along a frontier at a certain depth. Exploration is limited to $b$ nodes at each depth, where $b$ is selected as a hyperparameter that trades search accuracy (in terms of model score) for speed. From depth 0 composed of just root node $\varepsilon$ until a maximum depth limit $Y$ is reached, or another stopping criteria is reached (Klein et al., 2017), beam search progressively scores all children of the nodes at the current depth and prunes all generated nodes except for the top scoring $b$ nodes. Beam search scores each node with only the current cost $\hat{v}(p) = c(p)$, and, in the case of neural machine translation, setting $f(p) = 0$. Discarding the future cost biases search towards nodes with high scores without regard to whether they lead to a good path to the terminal state. Thus, a large space of potentially good translations with low initial scores is never explored. Examples include re-orderings that place high-entropy words at the start of the sentence or shorter sentence constructions (e.g. "Help me" vs. "Lend me a hand").

Interestingly, autoregressive models tend to overestimate the probability of short and ungrammatical translations that do not translate the entirety of the source sentence, which are pruned by this scoring heuristic (Stahlberg & Byrne, 2019; Holtzman et al., 2019). Thus, while $b$ may be set low to increase speed, it is often set low to improve translation *quality*. However, we believe that model changes evaluated with beam search's biases may be obscured. We seek to propose modeling improvements to mitigate degenerate solutions, such that search quality and model quality are aligned.

## 2.2 MONTE CARLO TREE SEARCH

In Monte Carlo tree search (Coulom, 2006, MCTS), the Monte Carlo method replaces the heuristically driven measure of value $\hat{v}(p)$ for a node $p$ with an expected-outcome model based on random game playouts. For instance, given a node $p$ representing a state of a game, one can assign a value to $p$ by randomly playing from that state a certain number of times and computing the average score obtained from the playouts. Thus, no burden is placed on the form of the objective function, allowing the definition of arbitrary complex objectives (e.g. winner of a chess game). Additionally, as each playout yields a possible terminal state, both current $c(p)$ and future costs $f(p)$ are naturally embedded within the obtained estimate. Unlike beam search and A$^*$ search, where the search direction is determined by value $\hat{v}(p)$, MCTS diversifies the search space by allocating budget to less explored areas in the search space (Kocsis & Szepesvári, 2006), and continually refines value estimates.

## 3 ADAPTIVE TREE SEARCH FOR TEXT GENERATION

While MCTS (Coulom, 2006) can be directly applied to decode arbitrary translation objectives, the heuristics defined in MCTS are optimised for environments where the computational cost of the scoring function $s(\boldsymbol{x}, \boldsymbol{y})$ is low. For instance, the playout heuristic $\hat{v}$ in the game of go (Silver et al.,

2016) runs hundreds of thousands of playouts, which can be computed in less than a second. In neural text generation the scoring function frequently requires the computation of a neural network forward step, such as a log-probability computed using an autoregressive model, rendering these practices prohibitive. One option is to rely on a heuristic to generate samples to train a neural network that estimates $\hat{v}$ (Leblond et al., 2021). However this has been shown to be challenging as model scores are difficult to estimate. Instead, we describe a variant of MCTS optimised on decoding text generation.

## 3.1 Deterministic Playout Heuristic

We start by establishing our playout function $\hat{v}(p)$, which is used as an initializer for node values. While the Monte Carlo method is effective at accurately estimating the value of a given sequence by performing multiple random playouts, such practice is infeasible as each of the playouts needs to be scored using the scoring function $s(\boldsymbol{x}, \boldsymbol{y})$, which we expect will be prohibitively expensive. Furthermore, chances that grammatical translations are sampled using this process are extremely low due to the sparsity of high-quality translations in $\mathcal{Y}$. Thus, rather than multiple random playouts, we estimate the cost of a given node using a single *informed* playout, which is guided by greedy decoding using an autoregressive model. Therefore, given a node $p$ with prefix $y^{(p)}$, we compute $\hat{v}(p)$ by recursively selecting the highest probability token $y$ according to an autoregressive model $p(y \mid y_1^{(p)}, \ldots, y_i^{(p)}, \boldsymbol{x})$, and scoring the translation using the objective function $s(\boldsymbol{x}, \boldsymbol{y})$.

This also implies our approach does not employ the Monte Carlo estimates and the decoding method is fully deterministic. The progression of the value estimates relies only on the refinement of the initial value estimates performed as the tree expands introduced in MCTS. Therefore, we will refer to our algorithm as an **adaptive tree search** (ATS) algorithm.

## 3.2 Adaptive Tree Search with a modified UCT Criteria

ATS operates on search trees instead of weighted graphs. A search tree covers the space of all possible full and partial translations $\mathcal{Y}^*$, and each node encodes a particular sequence $\boldsymbol{y}^{(p)}$. Nodes have $|\Sigma| + 1$ children, each appending a new word $y \in \Sigma \cup \{\text{EOS}\}$ to the sequence $\boldsymbol{y}^{(p)}$. We denote the child resulting from concatenating $y$ to $p$ as $p \circ y$. The child of a node that selects the EOS symbol is a terminal node, which is associated with a element in $\Sigma^*$, and therefore, can be scored using the objective $s(\boldsymbol{x}, \boldsymbol{y})$. Each node stores the number of visits $n^{(p)}$ and its current value estimate $v^{(p)}$, which can be reassigned during search. Nodes that have not been inserted in the tree have $n^{(p)} = 0$ and no estimate for $v^{(p)}$.

Search starts with the root node $\varepsilon$ with visit count $n^{(\varepsilon)} = 1$ and value $v^{(\varepsilon)} = \hat{v}(\varepsilon)$, which corresponds to the score obtained by translating $\boldsymbol{x}$ with greedy decoding. Afterwards the tree expands in an iterative manner, where each iteration expands the search tree and updates its statistics.

Similar to the selection and expansion steps in ATS, we traverse the instantiated tree, starting from $\varepsilon$ on the basis of the current estimated values $v$, together with confidence about the quality of the estimates. We recursively traverse the tree and select the child $p \circ y$ with the highest score according to the continuous upper confidence tree criterion (Auger et al., 2013):

$$\text{UCT}(p, y) = \bar{v}(p \circ y) + C \frac{\sqrt{n^{(p)}}}{1 + n^{(p \circ y)}} \pi(y \mid \boldsymbol{y}^{(p)}), \tag{1}$$

where $C$ is a hyperparameter weighting two terms. The first term $\bar{v}(p \circ y)$ encourages exploitation of nodes with known high value. The second term $\frac{\sqrt{n^{(p)}}}{1+n^{(p \circ y)}} \pi(y \mid \boldsymbol{y}^{(p)})$ encourages the algorithm to explore nodes with low visit counts more thoroughly. Here, we specify the policy as the log-probability obtained from an autoregressive model $\pi(y \mid \boldsymbol{y}^{(p)}) = p(y \mid \boldsymbol{y}^{(p)}, \boldsymbol{x})$. The value of a node is determined by its current estimate $\bar{v}(p \circ y) = v^{(p \circ y)}$ if $n^{(p \circ y)} > 0$. For nodes not yet inserted in the tree, which have no value estimates, we compute an estimated value as:

$$y^* = \underset{\forall y \in \Sigma, n^{(p \circ y)} > 0}{\arg\max} v^{(p \circ y)}$$

$$\bar{v}^{(p \circ y)} = v(p \circ y^*) \frac{\pi(y \mid \boldsymbol{y}^{(p)})}{\pi(y* \mid \boldsymbol{y}^{(p)})}, \tag{2}$$

where we estimate $\bar{v}^{(p \circ y)}$ by assuming that the ratio between policies $\pi$ between $p \circ y$ and the highest value node $p \circ y^*$ is the same as the ratio between their values $\bar{v}$.

The traversal terminates when a node with $n^{(p)} = 0$ or a terminal node is reached. In the former case, the node $p$ is inserted into the tree, setting its visit count $n^{(p)} = 1$ and estimating its value $v^{(p)} = \hat{v}(p)$ as done in the simulation step in MCTS. We note here an important difference to many other formulations of MCTS, where selection terminates at leaf nodes (node where all $\Sigma$ have $n^{(p)} > 0$), which is followed by the expansion step that inserts a new child prior to simulation. Expanding all children of a node is generally considered efficient in domains with a small $\Sigma$ and low playout cost $\hat{v}(p)$, and the standard MCTS algorithm does not attempt to optimise the subset that needs to be expanded. In the text domain, most words in the vocabulary are not applicable as they do not fit the context $\boldsymbol{y}^{(p)}$ and correspond to the content in the source sentence $\boldsymbol{x}$, and can be excluded using the value estimate described in Equation 2.

Next, we ascend from the selected node $p \circ y$, updating visit counts and value estimates:

$$n^{(p)} \leftarrow n^{(p)} + 1$$
$$v^{(p)} \leftarrow \max\{v^{(p)}, v^{(p \circ y)}\},$$

where each parent $p$ increases its visit count $n^{(p)}$ and updates its value estimate $v^{(p)}$ to the child's value if a new best translation is found. Thus, $v^{(p)}$ represents the best translation obtained in the subtree represented by $p$. Starting from the score obtained using greedy decoding when $v^{(p)}$ is initialised, each new traversal that passes through $p$ has a chance to refine this initial estimate with the newly found translation.

### 3.3 Beam Adaptive Tree Search

A standard way to guarantee progression in MCTS is to run an instance of MCTS per word. Here, we would run an ATS instance $\text{ATS}(\varepsilon, k)$ with $k$ iterations starting from root $\varepsilon$. Then, we set $\varepsilon \leftarrow \varepsilon \circ y_1^*$, where $y_1^* = \arg\max_{y \in \Sigma} v^{(\varepsilon \circ y)}$ is the child with the highest value estimate and repeat this process until $y_i^* = \text{EOS}$.

However, it has been found that in text generation tasks restricting search to a set of high value nodes rather than a single one allows such games to be solved at a faster rate (Baier & Winands, 2012). Thus, we modify our selection step as follows:

$$\text{UCT}_{constrained}(p, y, d_{min}) = \begin{cases} \text{UCT}(p, y) & d^{(p \circ y)} > d_{min} \\ -\infty & \text{otherwise} \end{cases} \tag{3}$$

where $d^{(p \circ y)}$ is initialised as $\ell(p \circ y)$, a function that counts the number of edges required to reach the root node from $(p \circ y)$. Then, the following update rule is added to ensure that the $d^{(p)}$ stores that depth of the deepest node achievable from $p$:

$$d^{(p)} \leftarrow \max\{d^{(p)}, d^{(p \circ y)}\},$$

where we update each node so that $d^{(p)}$ stores the value of the deepest node that is accessible from $p$. Thus, in Equation 3, condition $d^{(p \circ y)} > d_{min}$ tests whether $p \circ y$ contains a node deeper than $d_{min}$.

We define $\text{BATS}(\varepsilon, k)$ as a Beam ATS instance that runs ATS starting from the root node $\varepsilon$ with the selection criteria $\text{UCT}_{constrained}$ with $d_{min} = 0$ and gradually increasing $d_{min}$ by 1 every $k$ iterations. Search stops when no node satisfies $d^{(p \circ y)} > d_{min}$ or until a maximum depth $d_{max}$.

## 4 Objectives

As decoding with a decoder on vanilla autoregressive models is unlikely to yield translations with quality superior to beam search, as the beam search bias is essential to overcoming the calibration issues in these models, we propose modeling improvements in order to address these shortcomings.

### 4.1 Max Rank

Decoding in autoregressive models generally optimises a normalised log probability (Wu et al., 2016), $(\log p(\boldsymbol{y} \mid \boldsymbol{x}))(\frac{6}{5+|\boldsymbol{y}|})^\alpha$, which combines the sum of the token level log-probabilities (when $\alpha = 0$)

and a length-based adjustment, which approximates the mean of the log-probabilities as the length $|\boldsymbol{y}|$ grows (when $\alpha = 1$).

Similar to normalised log-probabilities, we consider a metric that characterizes translation by their minimum token level log-probability. The intuition here is that the quality of the translation is represented by the worst decision made in the sequence. In practice, many degenerate cases in autoregressive models are created by making a single bad decision, such as generating a EOS token prematurely or omitting translations, which can be understood in terms of uniform information density (Meister et al., 2020).

However, the issue with the minimum of the token level log-probability is that log-probability ranges tend to vary depending on the context and number of translation options that are available. Thus, they are not very reflective on the quality of the choice made as the best choice at a given timestamp could still be the worst decision in the sequence. Instead, we optimise the normalised rank:

$$r(y_i \mid y_1, \ldots, y_{i-1}, \boldsymbol{x}) = \frac{\sum_{y \in \Sigma} \delta(p(y_i \mid y_1, \ldots, y_{i-1}, \boldsymbol{x}) > p(y \mid y_1, \ldots, y_{i-1}, \boldsymbol{x}))}{|\Sigma|}$$

where we count the number of actions in $\Sigma$ with lower log-probability than $y_i$. By using the rank $r$ instead of the log-probability $p$, we can compare values within the same range at the cost of a loss in relative precision. Thus, we name our metric max rank (MR), which is computed as follows:

$$\text{MR}(\boldsymbol{x}, \boldsymbol{y}) = \max_{i=1}^{|\boldsymbol{y}|} \log r(y_i \mid y_1, \ldots, y_{i-1}, \boldsymbol{x}).$$

Finally, unlike the mean and sum of log-probabilities, the max of a sequence of log-probabilities is not autoregressive, so beam search is not applicable.

## 4.2 NOISY CHANNEL MODEL

The noisy channel model uses the Bayes rule decomposition in order to decompose the probability of a sentence $p(\boldsymbol{y}|\boldsymbol{x})$ into $p(\boldsymbol{y}|\boldsymbol{x}) = \frac{p(\boldsymbol{x}|\boldsymbol{y})p(\boldsymbol{y})}{p(\boldsymbol{x})}$ where the channel model $p(\boldsymbol{x}|\boldsymbol{y})$ can be trained as translation model trained in the reverse direction and $p(\boldsymbol{y})$ is a language model. Finally, the prior $p(\boldsymbol{x})$ can be ignored in the context of a maximization problem. Since the reverse model is not autoregressive in the space $\Sigma^*$, it can bypass many of the degenerative cases in autoregressive models.

## 4.3 MINIMUM RISK TRAINED AUTOREGRESSIVE MODELS

In order to show that BATS can be an attractive alternative to beam search under autoregressive models, we need to improve the model so that the search bias is no longer as crucial to preserving the translation quality of the generated text. To this end, we fine-tune our models using minimum risk training (Shen et al., 2016, MRT). The MRT training objective is designed to minimize the empirical risk $r(\mathbf{y}, \mathbf{y}')$ by minimizing it in a subset of $\Sigma^*$ obtained by sampling $n$ translations. This allows the model to mitigate degenerate cases caused by optimising the likelihood objective by fine-tuning the model on downstream metrics, such as BLEU (Papineni et al., 2002).

# 5 EXPERIMENTS

## 5.1 SETUP

We conduct our experiments on the Chinese–English and Pashto–English tasks from WMT2020 (Barrault et al., 2020), and German–English from WMT2014 (Bojar et al., 2014), following the same training, development and test splits. Our autoregressive model transformer baseline uses the multi-query attention model (Shazeer, 2019). It uses the standard architecture with 6 encoder and decoder layers with 512 hidden units, 2048 sized tied embeddings for both source and target word projections and 8 attention heads. We tokenize the data with byte-pair encoding (Sennrich et al., 2016) with 32K merges and set a maximum sentence size of $Y = 128$. Translation quality evaluation is performed using sacreBLEU (Post, 2018). We choose the checkpoint that yields the highest BLEU in the validation set using beam search with the normalisation constant $\alpha = 0.8$ and beam size 6. We also compare a variant fine-tuned using MRT according to the procedure described in Appendix A.3.

For the noisy channel model, we train the channel model by simply swapping the translation direction of the autoregressive model with the same hyperparameters. For the language model prior, we employ the TransformerXL architecture (Dai et al., 2019) trained on 1 billion words (Chelba et al., 2013). On non-autoregressive objectives, we use beam search as a proxy to generate translations candidates which are rescored using the non-autoregressive metric (Yee et al., 2019; Yu et al., 2020a). For BATS, we simply set hyperparameter $C = 1$. While optimising $C$ could lead to more efficient optimisation of the model score, our goal is to study how model scores are correlated with translation quality in different objectives.

The translation budget (beam size for beam search and iterations for BATS) is swept by doubling its value starting from 1 to 256. We combine different objective components under a log-linear model, where the weights of the components are tuned with MERT (Och, 2003). In order for model scores to be comparable, all weights are tuned on a pool of 256 translations using beam search.[2]

## 5.2 RESULTS

Table 1 illustrates the translation quality results using BATS and beam search on the normalised autoregressive model baseline and the optimal model (Column "System") for each language pair (Column "Language Pair"). We perform a grid search over the following decisions: (1) Whether to use Max Rank (Column "MR"), (2) Whether to use the Noisy Channel Model (Column "NC"), (3) Whether to tune the autoregressive model using MRT (Column "MRT"), (4) Whether to use beam search or BATS (5) The translation budget of each decoder. The combination with the highest BLEU on the validation set is used to decode on the test set and the BLEU scores obtained using beam search and Bats are reported (Columns "Beam Search" and "BATS").[3] We observe that for all pairs decoding with BATS can yield gains over decoding with beam search when using the combination of objectives with the highest BLEU on the validation set. As expected, due to the search bias in beam search, it does comparatively better on some language pairs, such as Chinese-English and Pashto-English.

In terms of modeling, we note that our proposed metric, Max Rank, and the MRT method combined yield the best results for Mandarin and German. The noisy channel model only yields improvements in Pashto-English, when used with Max Rank and MRT as the training data is small (500k parallel sentences) and the model relies on the large sized language model to provide accurate predictions.

| Language Pair | System | MR | NC | MRT | Beam Search | BATS |
|---|---|---|---|---|---|---|
| Chinese–English | Baseline | No | No | No | **24.7** | 24.2 |
| | Optimal | Yes | No | Yes | 28.6 | **29.0** |
| Pashto–English | Baseline | No | No | No | **7.5** | 7.3 |
| | Optimal | Yes | Yes | Yes | 8.1 | **9.1** |
| German–English | Baseline | No | No | No | 30.0 | 30.0 |
| | Optimal | Yes | No | Yes | 30.3 | **31.1** |

Table 1: Comparison between the BLEU scores obtained using beam search and BATS. Pairs of rows describes the results obtained using the vanilla autoregressive model with normalisation and the best combination of models (Max Rank, Noisy Channel and Minimum Risk Training), tuned on the validation set. The translation budget is also tuned for each method for maximum BLEU.

## 5.3 BEAM SEARCH AND BATS

We now provide a more in-depth analysis on the Chinese-English language pair, where we believe results are more informative. Table 2 illustrates the results obtained using some models that were explored in our grid search (Column $s(\boldsymbol{x}, \boldsymbol{y})$). For each model, we illustrate the best BLEU obtained on the test set (Column "BLEU"), the beam size (Column "Beam") or number of iterations (Column "Iter"), where the best BLEU was obtained on the validation set and the percentage improvement between results obtained using beam search and BATS (Column "Delta"). Finally, autoregressive and non-autoregressive models are marked with $AR$ with $NAR$, respectively. It is also important to

---

[2]Tuning $\lambda$ using BATS to genererate candidates yields similar weights.
[3]The optimal models with the highest BLEU on the validation for Beam Search and BATS are the same.

refer that the number of iterations in BATS and beam in beam search are not comparable point-wise, instead we analyse the overall behavior of the model score and BLEU curves.

**When to use Beam Search?** In the models where beam search outperforms BATS(Rows "Log Probability$^{AR}$ ($\alpha = 0$)" and "Log Probability$^{AR}$ ($\alpha = 0.8$)"), we notice that the search budget in both cases is always low. Figure 1 plots the evolution of the BLEU (Top) and model score (Bottom) of both decoders as translation budget is increased. For the normalised log-probability (Column "Log Probability ($\alpha = 0.8$)"), the model scores obtained for BLEU search and BATS are very similar. However, we an observe a large gap between the BLEU scores obtained at the same model score. This shows that the beam search bias is filtering the set of candidates in beam search so that they are of higher quality, even though the model score is failing to discriminate them. Additionally, we observe the standard BLEU curve (Stahlberg & Byrne, 2019), where after a set of initial iterations BLEU deteriorates rapidly, which renders elaborate decoding mechanisms unneeded. Furthermore, BATS and other MCTS-based methods are not ideal for low search budget scenarios as they depend on high node visit counts to accumulate enough statistics to make informed decisions. We see in all objectives that model scores for BATS do not outperform Beam Search until a large budget is allocated. In conclusion, if the model cannot support high search budgets, beam search is the preferred alternative, especially if the modeling issues can be addressed with the search bias.

**When to use BATS?** With the addition of the noisy channel model (Row "Log Probability ($\alpha = 0.8$) + NC$^{NAR}$) and Max Rank (Row "Log Probability ($\alpha = 0.8$) + MR$^{NAR}$"), the search budget until BLEU deteriorates for both decoders is increased. Here, we observe that BATS is the decoding option that yields higher BLEU and that the delta is higher when BATS can employ more iterations.

Figure 1 shows that the evolution of BLEU is considerably more stable with the noisy channel model (Column "Log Probability ($\alpha = 0.8$) + NC") and when Max Rank is applied (Column "Log Probability ($\alpha = 0.8$) + MR"). For these non-autoregressive models BATS yields significantly better model scores than reranking. More interestingly, comparing the behavior of Beam Search and BATS, similar model scores between the two methods do not yield similar BLEU scores. In the noisy channel model, we observe that at 16 iterations and beam size 16, both decoders yield similar model scores and BLEU scores. However, as the model score increases beyond that point, the degeneration in beam search as the score increases is significant, while BATS observes almost no deterioration. This suggests that the beam search bias has a negative impact in translation quality at high values of $b$ by filtering good translations from the search space. Using Max Rank, we observe that not only BATS can achieve considerably higher model scores as it is optimising the non-autoregressive model directly, it also yields considerably higher BLEU scores. In beam search, BLEU stops improving at 32 iterations even though model score keeps increasing due to the search bias.

Finally, we observe that the search bias issue is also present when using beam search to decode from an autoregressive model fine-tuned with MRT ( Table 2, Row "Log Probability MRT ($\alpha = 0.8$)$^{AR}$"). Figure 1 shows that once MRT is applied (Column "Log Probability MRT ($\alpha = 0.8$)"), not only BATS can find significantly better model scores, but they are are associated with better BLEU scores. Additionally, beam search stagnates after 8 iterations.

In conclusion, as translation models become more robust, there is a growing need of better decoding mechanisms, such as BATS, in order to maximize translation quality. We perceive that in both autoregressive and non-autoregressive models, there is a limit to both model and BLEU scores that can be obtained using beam search, which is partially attributed to the fact that its search is strongly biased due to the lack of future costs.

We believe that future research will drive models to extents where translation quality nearly perfectly matches with model scores. In an oracle setup, where the objective is the sentence level BLEU by peeking at the reference (Row "Oracle BLEU"), we hypothesise that the delta between beam search and BATS would grow vastly, and observe that a delta of $39.19\%$.

## 6    CONCLUSION

This paper proposes an adaptive tree search algorithm designed to optimise arbitrary metrics. It uses rollouts from an auxiliary autoregressive model to obtain estimates for the value estimates of internal nodes. This allows the decoder to optimise arbitrary objectives and avoid the search biases of

| $s(\boldsymbol{x}, \boldsymbol{y})$ | Beam Search BLEU | Beam | BATS BLEU | Iter | Delta |
|---|---|---|---|---|---|
| Log Probability$(\alpha = 0)^{AR}$ | **24.7** | 2 | 24.2 | 1 | -2.0% |
| Log Probability$(\alpha = 0.8)^{AR}$ | **25.0** | 8 | 24.7 | 8 | -1.2% |
| Log Probability $(\alpha = 0.8)$ + NC$^{NAR}$ | 25.2 | 16 | **25.4** | 32 | 0.6% |
| Log Probability $(\alpha = 0.8)$ + MR$^{NAR}$ | 26.8 | 32 | **27.4** | 128 | 2.5% |
| Log Probability $(\alpha = 0.8)$ + MR + NC$^{NAR}$ | 26.7 | 32 | **27.4** | 128 | 2.6% |
| Log Probability MRT $(\alpha = 0.8)^{AR}$ | 28.2 | 64 | **28.7** | 256 | 1.8% |
| Log Probability MRT $(\alpha = 0.8)$ +$MR^{NAR}$ | 28.6 | 8 | **29.0** | 256 | 1.7% |
| Oracle BLEU$^{NAR}$ | 38.6 | 256 | **53.7** | 256 | 39.2% |

Table 2: Comparison between BS and MCTS on the WMT2020 Chinese-English test set. Pairs of cells denote BLEU scores and the iteration budget achieving the best BLEU on the validation set.

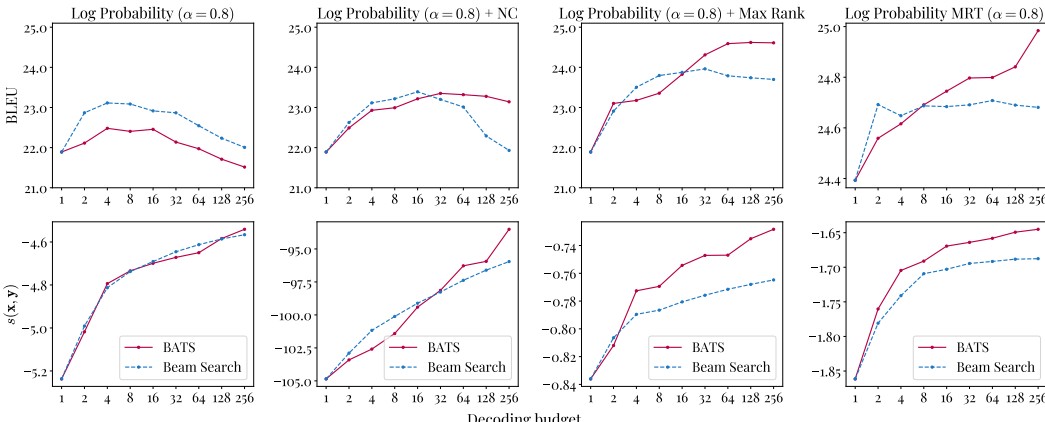

Figure 1: Comparison of BATS and beam search over different translation budgets under different translation objectives on the WMT2020 Chinese–English test set. Each column illustrates the BLEU (top) and the model score (bottom) obtained using the two decoders on a different objective.

manually defined heuristics, such as partial translation probability in autoregressive models. BATS is particularly useful when models are robust enough to allow for a higher search budget. As many existing objectives are found to be poorly correlated with translation quality, we propose a new metric named max rank and use existing methods, such as the noisy channel model and minimum error rate training, to address the failure modes of vanilla autoregressive models. Results on three language pairs are favourable to BATS when using our proposed augmentations of the autoregressive model. Additionally, we find that the gap in translation quality between beam search and BATS increases as more robust models are employed. More importantly, we observe that the search bias prevents beam search from achieving high-quality translations as it filters good translations that are unfavoured by the search heuristic. Thereby, the model score increases, but BLEU decreases. This shows that beam search limits the potential of many models by establishing translation quality ceilings unrelated to the robustness of the model, but to the topology of the search space they establish. This suggests that as scientific progress drives more robust models, exploring more robust decoding methods, such as BATS, is fundamental for advancing the field of text generation.

ACKNOWLEDGMENTS

We thank Rémi Leblond and Phil Blunsom for useful feedback throughout the different stages of this project.

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

# A APPENDIX

## A.1 BATS VS. ATS

The advantage of the beam variant of MCTS (Baier & Winands, 2012) is that the search algorithm does not have to commit to a single branch every $k$ iterations. As often occurs during translations, multiple valid translation exist and correspond to different branches in the tree and only after furthering the search tree can the optimal translation be filtered out. In ATS, once the EOS is chosen, search ends immediately, with no chance for the decoder to explore other branches, which accentuate issues where degenerate solutions are chosen (e.g. prematurely ending a translation). A comparison between ATS and BATS on our WMT2020 Chinese-English validation set using autoregressive models is shown in Table 3, where we observe that ATS has both a bias towards degenerate cases and yields worse model scores and BLEU.

|  | BATS | | ATS | |
|---|---|---|---|---|
| Iter | $s(\boldsymbol{x}, \boldsymbol{y})$ | BLEU | $s(\boldsymbol{x}, \boldsymbol{y})$ | BLEU |
| 1 | -5.2 | 21.9 | -5.23 | **21.9** |
| 2 | -5.00 | 22.9 | -4.92 | 21.3 |
| 4 | -4.81 | **23.1** | -4.90 | 20.9 |
| 8 | -4.74 | 23.1 | -4.80 | 20.8 |
| 16 | -4.69 | 22.9 | -4.78 | 20.7 |
| 32 | -4.65 | 22.9 | -4.75 | 20.7 |
| 64 | -4.61 | 22.5 | -4.73 | 20.4 |
| 128 | -4.59 | 22.2 | -4.72 | 20.2 |
| 256 | -4.57 | 21.7 | -4.72 | 20.0 |

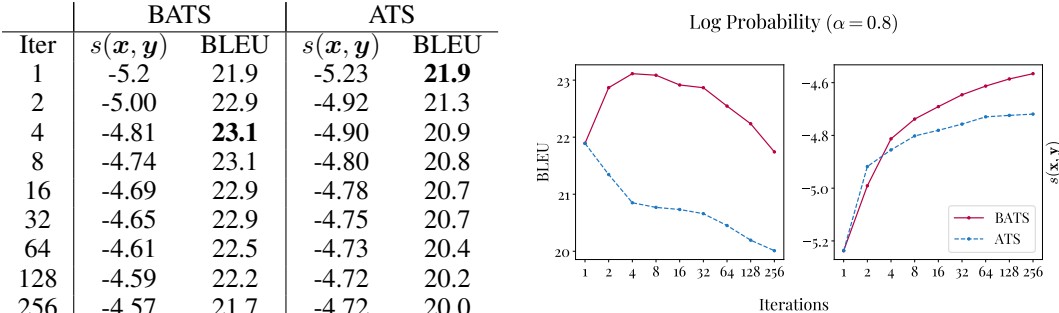

Table 3: Comparison between BATS and ATS on our WMT2020 Chinese-English validation set, with Log Probability ($\alpha = 0.8$) as the search objective.

## A.2 MIN PROB VS. MAX RANK

The most straight-forward approach to select the worst decision in a translation is to select the lowest log-probability in the sentence. However, log-probabilities are not a good indicator of whether a decision is good or bad as some word translations are inherently low probability (words with many valid translations). Thus, we decided to use the maximum rank instead. Table 4 provides a comparison between the Min Prob (Row "Log Probability ($\alpha = 0.8$) + MP") and the Max Rank objective (Row "Log Probability ($\alpha = 0.8$) + MR"). We observe that while Min Prob yields a relatively small improvement, it is significantly smaller than Max Rank's improvement over the baseline (Row "Log Probability($\alpha = 0.8$)"). Additionally, it clearly does not address the degenerate solutions problem in autoregressive models.

| | Beam Search | | BATS | | |
|---|---|---|---|---|---|
| $s(\boldsymbol{x}, \boldsymbol{y})$ | BLEU | Beam | BLEU | Iter | Delta |
| Log Probability($\alpha = 0.8$)$^{AR}$ | **25.0** | 8 | 24.6 | 8 | -1.2% |
| Log Probability ($\alpha = 0.8$) + MP$^{NAR}$ | **24.8** | 8 | 24.7 | 16 | -0.3% |
| Log Probability ($\alpha = 0.8$) + MR$^{NAR}$ | 26.8 | 32 | **27.4** | 128 | 2.5% |

Table 4: Comparison between Min Prob and Max Rank in our WMT2020 Chinese-English test set. The number of iterations is tuned on the validation set.

A.3    MINIMUM RISK TRAINING WITH CHRF AND BLEU

While BLEU (Papineni et al., 2002) is the downstream translation metric used in most MT evaluations, its sentence level predictions tend to be sparse and inaccurate. Thereby, training quickly overfits before optimal translation quality is reached. With sentence level ChrF (Popović, 2015) training is more stable, and a better optimal translation quality can be obtained.

We use a sample size of $8$ translations per sentence, and these are generated via temperature sampling with temperature 0.8. Finally, we set the risk $r(\mathbf{y}, \mathbf{y}') = -\frac{1}{2}\text{BLEU}(\mathbf{y}, \mathbf{y}') - \frac{1}{2}\text{ChrF}(\mathbf{y}, \mathbf{y}')$, where $\text{BLEU}(\cdot)$ is the sentence level sacreBLEU (Post, 2018) and $\text{ChrF}(\cdot)$ is the sentence level ChrF (Popović, 2015). We used the average of BLEU and ChrF because BLEU was designed as a corpus level metric, and ChrF provides a better estimate of translation quality with sparser matches against a reference for a single sentence.

Table 5 compares the results obtained using only bleu($\cdot$), chrf($\cdot$) and their combinations, using Beam Search with beam 6, on the validation set in the WMT2020 Chinese-English dataset. Here, we can observe that a combination of both scores yields the optimal translation quality.

| bleu($\cdot$) | chrf($\cdot$) | BLEU |
|:---:|:---:|:---:|
| - | - | 23.1 |
| 1 | 0 | 23.8 |
| 0 | 1 | 24.4 |
| $\frac{1}{2}$ | $\frac{1}{2}$ | **24.7** |

Table 5: BLEU obtained on the WMT2020 Chinese-Englsih validation set using different metrics as risk $r(\mathbf{y}, \mathbf{y}')$. Columns "bleu($\cdot$)" and "chrf($\cdot$)", denote the weights applied and Column "BLEU" denote the BLEU obtained. The first row illustrates the BLEU score obtained prior to MRT.

A.4    COMPUTATIONAL COST OF BATS

For Beam Search with beam size $b$ and a max sentence length $Y$, beam search requires $b \times Y \times A$ operations, where $A$ is a transformer+softmax block. Additionally, for reranking, the model score $s(\boldsymbol{x}, \boldsymbol{y})$ needs to be computed for each of the $b$ translations and has a computational cost of $B$. Thus, the total cost is $b \times (Y \times A + B)$. In BATS, computation is proportional to the number of expanded nodes, when the playout heuristic is applied. Each playout requires the computation of greedy decoding, followed by the scoring function $s(\boldsymbol{x}, \boldsymbol{y})$. Thus, each rollout requires $Y \times A + B$ computations. Finally, all nodes that follow the path used in greedy decoding will have the same value $v$, which means that the first node that is expanded, which corresponds to this path will have no cost, with the exception of the root node. Thus, the cost of BATS is $(1 + z) \times (Y \times A + B)$, where $z$ is the number of non-root nodes expanded more than once. As $Y \times A + B$ is a common denominator for both methods, we define it as the a computational unit.

Table 6 shows the results obtained for max rank objective (row "Log Probability ($\alpha = 0.8$) $+MR$" in Table 2). The "Cost" column represents the number of computational units, and we observe that BATS is considerably more expensive to run than Beam Search. However, observe the model scores (Column "Score"), we notice that Beam Search gradually decreases the rate at which model score gains are observed (Column "Gain") even though the cost doubles at each row. For BATS, we observe that while the cost is extremely high initially (73.049 at 2 iterations), the cost increases at a linear rate with the number of iterations. More importantly, we notice considerable gains even with high numbers of iterations (11.82 at 128 iterations). Finally, at beam $256$, we notice that the cost of Beam Search is comparable to the cost of running 16 BATS iterations, which achieves similar model scores.

It is important to also refer that cost efficiency is not the goal of this work, but to expose the need for better decoders that are devoid of the beam search bias. Many improvements to MCTS-based methods can be made to improve efficiency, such as training value and policy networks iteratively (Silver et al., 2016).

| Beam/Iter | Beam Search | | | BATS | | |
|---|---|---|---|---|---|---|
| | Cost | $s(\boldsymbol{x}, \boldsymbol{y})$ | Gain | Cost | $s(\boldsymbol{x}, \boldsymbol{y})$ | Gain |
| 1 | 1 | -0.84 | - | 1 | -0.84 | - |
| 2 | 2 | -0.81 | 0.0296 | 73.0 | -0.81 | 0.0240 |
| 4 | 4 | -0.79 | 0.0169 | 146.4 | -0.77 | 0.0394 |
| 8 | 8 | -0.79 | 0.0030 | 217.5 | -0.77 | 0.0032 |
| 16 | 16 | -0.78 | 0.0059 | 315.7 | -0.75 | 0.0151 |
| 32 | 32 | -0.78 | 0.0048 | 421.9 | -0.74 | 0.0072 |
| 64 | 64 | -0.77 | 0.0043 | 584.3 | -0.74 | 0.0002 |
| 128 | 128 | -0.77 | 0.0035 | 641.9 | -0.73 | 0.0118 |
| 256 | 256 | -0.76 | 0.0032 | 846.3 | -0.73 | 0.0068 |

Table 6: Comparison between BS and MCTS in terms of computational cost using the normalised autoregressive model in the WMT2020 Chinese-English validation set. Cells denote the computational cost, model score obtained and the gain on score obtained relative to the row above.

## A.5 EXAMPLE TRANSLATIONS AND SEARCH ERRORS

We compare the translated sentences using an a MRT tuned autoregressive model for Chinese-English, where BATS and beam search yield similar BLEU but where BATS achieves significantly lower model scores. Table 7 provides three example translations from the test set obtained using beam 64 for beam search (row "Beam Search") and 256 iterations for BATS (row "BATS"), which is the setup that obtained optimal results in the validation set. The first example shows that beam search tends to prolong sentences by using longer expressions ("many" vs. "there are also many") in order to get short term value gains, but lower overall score, also slightly shifting tone of the sentence. Figure 2 illustrates this issue, we observe that by using the expression "there are also many", it delays the generation of the word "technological" for three timestamps, leading to the higher score of $-0.78729$ (left path) compared to the alternative $1.20077$ (right path) in the same timestamp. While this score regularizes to $-1.58682$ once the word "technological" is generated, its likely that the alternative translation is pruned by beam search.

In the second example, we observe that beam search prefers to reorder the original sentence, so that higher scoring terms ("U.S. destroyer USS Decatur") in the sentence are inserted first. However, as one can observe from the final score, this decision is only favorable in the short term as the final score of the sentence is substantially lower than the translation found using BATS, which respects the order of the original sentence. In the last sentence, we observe that in the final portion of the translation, the decoder makes a set of individually high scoring decisions that lead to an ungrammatical translation as all grammatical options have been filtered from the beam.

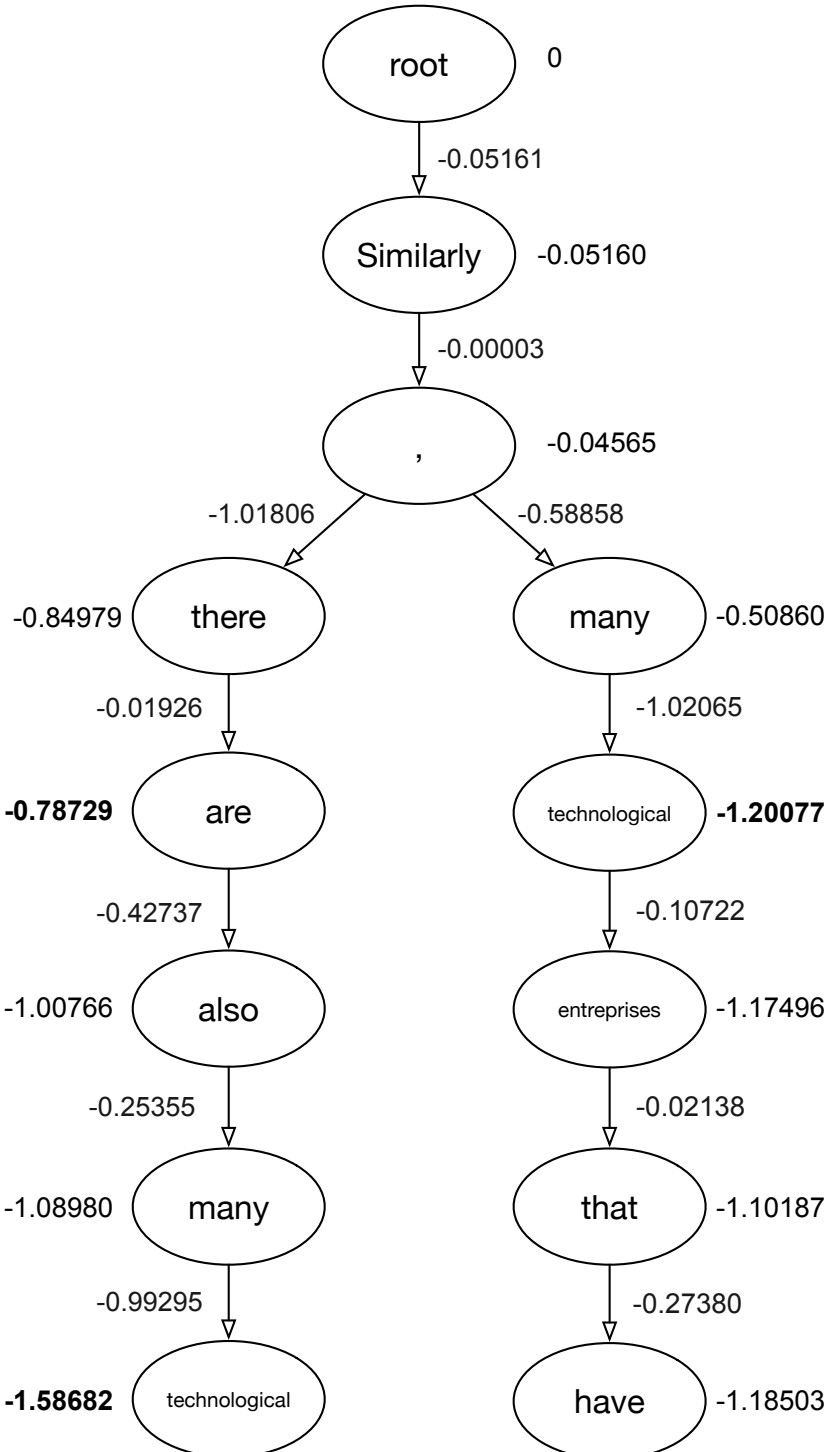

Figure 2: Illustration of the issue with the search bias in beam search where the decoder can delay the generation of low probability words, in this case the word "technological" in order to generate high initial scores. Edges scores correspond to the token level log probability $\log p(y_i \mid y_1, \ldots, y_{i-1}, \boldsymbol{x})$ and nodes scores correspond to the value that is assigned to the state using the normalised partial sum of probabilities $\frac{\sum_{j \leq i} \log p(y_j|y_1,\ldots,y_{j-1},\boldsymbol{x})}{(\frac{5+i}{6})^\alpha}$ with $\alpha = 0.8$. This example is obtained from the WMT2020 test set.

| | | $s(\boldsymbol{x}, \boldsymbol{y})$ |
|---|---|---|
| Source | 同样，也有不少科技企业在专利技术授权、专利技术使用等方面遭遇了不小的侵权风波。 | |
| Reference | Similarly, many technology companies have encountered numerous infringements crisis in patent technology licensing and patent technology use. | |
| Beam Search | Similarly, **there are also many** technological enterprises that have encountered great infringing waves in the licensing of patented technology and the use of patented technology. | -2.097 |
| BATS | Similarly, many technological enterprises have encountered great infringing waves in the licensing of patented technology and the use of patented technology. | -1.740 |
| Source | 据中央社30日综合外电报道，据要求匿名的美国官员透露，美国驱逐舰USS Decatur驶入了南沙群岛南薰礁（Gaven Reef）和赤瓜礁（Johnson Reef）12海里范围内。 | |
| Reference | According to the comprehensive foreign reports of the Central News Agency, the U.S. official who requested anonymity revealed that the United States Navy destroyer, USS Decatur, cruised into the 12 nautical mile territorial limit of Gaven Reef and Johnson Reef of the Nansha Islands. | |
| Beam Search | **U.S. destroyer USS Decatur** sailed into the Gaven Reef and Johnson Reef **12 nautical miles** (**12 nautical miles**) of the Southern Sand Islands, according to Central Intelligence Agency's Comprehensive Outreach News on 30. | -2.722 |
| BATS | According to Central News Agency's comprehensive external telecommunications report on 30 June, U.S. officials who requested anonymity, the USS Decatur sailed into the Gaven Reef and Johnson Reef of the Southern Sand Islands within 12 nautical miles. | -2.288 |
| Source | 意大利疑欧派政府的目标是未来三年预算赤字相当于国内生产总值(GDP)的2.4％，这表明仅管面临减赤要求仍未有债务削减 | |
| Reference | The aim of Italian Eurosceptic government was that the budget deficit was equivalent to 2.4% of gross domestic product (GDP) in the next three years, suggesting that there was no debt reduction despite deficit reduction requirements. | |
| Beam search | The Italian Euroskeptic government's goal is to have a budget deficit equivalent to 2.4% of gross domestic product over the next three years, **suggesting that only facing deficit reduction requirements has not yet been debt reduction.** | -1.647 |
| BATS | The Italian Euroskeptic government's goal is to have a budget deficit equivalent to 2.4% of gross domestic product over the next three years, indicating only that there is no debt reduction required to reduce the deficit. | -1.542 |

Table 7: Examples of translation obtained using beam search and BATS on the Chinese-English test set using MRT tuned autoregressive models.

