# OpenReview forum: "Enabling Arbitrary Translation Objectives with Adaptive Tree Search"
_ICLR.cc/2022/Conference — ICLR 2022 Poster_

### Official Review · Reviewer_i2pz · 2021-11-02

**Correctness:** 3
**Technical Novelty And Significance:** 3
**Empirical Novelty And Significance:** 2
**Recommendation:** 6
**Confidence:** 4

**Main Review:**

Strengths:
+ The paper is well-structured and easy-to-follow even for someone who does not regularly follow latest research on MCTS.
+ The introduced depth control strategy of BATS is interesting and makes it easier to control for search budget compared to vanilla ATS.

Weaknesses:
+ My main complaint for this paper is that it did not cite an immediate related work of this paper (https://arxiv.org/pdf/2004.12527.pdf), which also applies MCTS to NMT and mentioned some similar ideas such as informed playouts, Although I realize that these two papers have some differences, I think it's more appropriate to cite it and clearly state the differences.
+ The gain brought by the proposed BATS method is overall very small (<=1 BLEU).
+ The evaluations miss some important details, and I have some doubts whether evaluations were presented in a way such that it unfairly favors BATS. For example, Table 1 only presented "the best combination of models... tuned on validation set", which I assume is tuned to optimize performance for BATS. Hence, this cannot be generalized as "BATS yields gains over beam search" -- I would rather see comparison of results from other configurations as well.
+ Since the author mentioned that the search budgets of BATS and beam search is not point-wise comparable, it'll be interesting to see a discussion on the computation time of BATS in the experiments, which is missing for now.

More detailed feedback:
+ More citation suggestions -- there were several SMT papers that should be cited in the discussion about beam search. Och et al. 2001 (https://aclanthology.org/W01-1408.pdf) should be cited for using A* search for MT. Koehn et al. 2003 (https://aclanthology.org/N03-1017.pdf) should be cited when discussing future cost of decoding for MT. Also, on MCTS, Kumagai et al. 2016 (https://aclanthology.org/W16-5502.pdf) is another paper that deployed MCTS to NLG.
+ Some missing details about evaluation: (1) I don't think the paper mentioned what non-autoregressive models were used for the experiments. (2) Using sacrebleu without including the signature is not helpful at all for future reproduction of results, and I would strongly urge the authors to include them in the paper. (3) Significance tests on translation quality differences will be helpful especially given the small diff in BLEU.
+ Some presentation suggestions: (1) It will be helpful to clarify the meaning of "true objective" and "s(x, y)" earlier on, as I have been wondering what objective is used for the search until section 4. (2) It took me a while to understand what "iteration" means in section 3.3. I would try to connect this with the notion of "traverse" in section 3.2.
+ Some minor stylistic comments: (1) Abstract: "than beam search has" -- remove "has"; (2) Page 2: I cannot parse the sentence starting with "For autoregressive models, we show that..."; (3) Page 3, first sentence of Section 3 "applied decode" -> "applied to decode"; (4) I cannot parse the sentence below equation 2; (5) Page 5, "update its value estimate $v^{(p)}$ to the child's value..." -- missing a "when" after this?

**Summary Of The Paper:**

This paper presents a novel algorithm called beam adaptive tree search (BATS) to enable the incorporation of search objectives that cannot be easily factorized. Unlike the regular Monte Carlo tree search algorithms that relies on a large number of playouts to update the value function, BATS relies on "informed playout", which is guided by the greedy decoding of the autoregressive models. To further constrain the search space, the paper also proposed a constrained node expansion criterion that gradually increases the lower limit of the node depth $d_{min}$, and allows for expanding $k$ nodes for each depth limit $d_{min}$.

Aside from the search algorithm, the paper also explored a couple of model changes to counteract the calibration issues, including two modified search objectives and using MRT-trained autoregressive models. Experiments show that while beam search performs better for the weaker models, BATS does improve the translation quality of the hypotheses when stronger MT models are used. Analysis on different beam size also show that unlike beam search, BATS can operate at large search budget without performance degradation under stronger models.

**Summary Of The Review:**

Nicely presented paper with some interesting proposals, but the novelty is not as significant as it seems due to some similar but missing related work. The small performance gain, together with doubts on evaluation soundness and/or real-world applicability, would probably also limit its impact.

---

> ### Author Response · Authors · 2021-11-19
> **Rebuttal (cont)**
>
> Regarding Table 1, results were optimized separately for beam search and BATS. There is a footnote stressing that:
> The optimal models with the highest BLEU on the validation for Beam Search and BATS are the same.
>
> Which means the set of components that were ideal for Beam Search and BATS are the same, which is expected since they are only different in the thoroughness of search, so good models tend to be good regardless of how good search is. The translation budget was also tuned for each decoder. Finally, if we believe there is a slight bias towards beam search, as beam search is used during training as a stopping criteria, so the model checkpoint that is selected optimizes beam search.
>
> —
> Regarding the computational efficiency of the proposed method, there is a section in the appendix (A.4) that provides such an analysis (you are not forced to read it but we really appreciate it if you did and we can put it into the main paper you find this relevant), where you can see that beam search at 256 is 4 times faster than BATS with 256 iterations. However,  the cost of BATS (~315) with 16 iterations is similar to that of Beam Search with 256 and gets slightly higher model scores.
>
> Furthermore, it shows that the gains BATS gain from increasing the search budget grow steadily, whereas you get diminishing gains as you increase the beam of beam search. This is because BATS is allocating the budget more efficiently. We also explain in section 5.3 that you should only use BATS if you are concerned about maximizing translation quality at a cost of high computational budget. Finally, we think it's a bit unfair that our method is being penalized for being slow. To be clear, beam search is also slow if very large beams are used (e.g. 1024) and even slower if all the 1024 translation hypotheses have to be reranked for the noisy channel models for instance. We consider beam search fast because we generally use it with low beams because there is little incentive to increase the beam size substantially as translation quality plummets, but we show that there are ways to address this issue. Thus, this search problem is an issue that all decoding methods need to address not just BATS. From our analysis, it seems BATS is actually more efficient in this scenario as it obtains better model scores.
>
> —
> Regarding the rest of the comments, we agree and are grateful for the typo fixes and bibliography suggested.

---

> > ### Comment · Reviewer_i2pz · 2021-11-27
> > **Reviewer Comment After Response**
> >
> > Thanks to the authors for the detailed response. I think the authors' response revealed problems that I would strongly advise the authors to fix in the next draft.
> >
> > + **Novelty**: Thanks for pointing out the differences in the approach. I think the argument you made about the differences between your work and Parker and Chen (2020) is actually very helpful for the readers to understand the full context of the existing literature. I notice this paper is still not cited in the updated draft. I understand the defects in that paper, but I don't think that's why it should not be cited -- I think the more proper way is to cite it and clearly state the difference, maybe also comment on its defects from your perspective.
> >
> > + "More importantly, previous work never attempts to address the search bias issue in machine translation (BLEU score decreases as search budget increases), and we provide a solution to this problem (e.g. Max rank)." This is an unfair comment -- in fact, this is a very active research area in machine translation. The paper that the authors chose to cite (https://arxiv.org/pdf/2104.05336.pdf) is an example of such work. I can easily list a few more:
> >
> >   - https://aclanthology.org/W17-3202.pdf
> >   - https://aclanthology.org/W17-3207.pdf
> >   - https://aclanthology.org/W18-6322.pdf
> >   - https://aclanthology.org/2020.coling-main.398.pdf
> >   - https://aclanthology.org/2021.acl-long.22.pdf
> >
> > It's actually worth summarizing the insights in those papers as a paragraph in the related work section, together with the SMT citations I suggested above.
> >
> > + "Furthermore gains are consistent in 3 different languages, which accounts for statistical significance." When I'm talking about statistical significance, I'm specifically talking about Koehn 2004 (https://homepages.inf.ed.ac.uk/pkoehn/publications/bootstrap2004.pdf). Several papers have pointed out the importance of such test (see section 3.2 of https://aclanthology.org/2021.acl-long.566.pdf and section 5.3 of https://arxiv.org/pdf/2107.10821.pdf), and in my opinion this is especially important when the gap is small. Honestly I don't expect all the differences to be statistically significant in this paper -- that's fine, as there is limited amount of improvement you can expect from just fixing search -- but it's important to communicate this aspect of the result clearly for it to be taken seriously by the MT community, and also as guidance for future reproducibility.
> >
> > + **Computational Efficiency**: Thanks for the pointer into the Appendix. Almost all the reviewers mentioned this one, so maybe this is worth being moved into the main sections.
> >
> > Regarding the overall evaluation, I think the authors made a satisfactory response to my main complaint (the first point about novelty), and I do think it's an interesting paper overall. Hence, I will improve my overall evaluation to marginal accept.
> >
> > I do, again, *strongly* suggest the authors to consider my comment here and appropriately address them in the future draft.

---

> > > ### Author Response · Authors · 2021-11-29
> > > **Response**
> > >
> > > We deeply thank the reviewer for reading our response and providing additional input, and also improving the initial score. We will update the citations and run the statistical significance tests at least for the main experiments of the paper.
> > >
> > > One last thing we wish to clarify is the fact that we have used the wrong wording in "More importantly, previous work never attempts to address the search bias issue in machine translation (BLEU score decreases as search budget increases), and we provide a solution to this problem (e.g. Max rank).". Firstly, we don't imply that no-one has worked on search issues with beam search before. However, this particular issue, which is referred as search bias as in (https://aclanthology.org/D19-1331/), describes the fact that beam search makes it get higher BLEU scores almost by overlooking high scoring translations. They show that A* search can actually achieve a much higher score than beam search, but with a near zero BLEU score. Of course, A* search is only applicable to small domains, and non-normalised auto-regressive models, since most translations underperform very short sentences, but our BATS decoder is similar but more scalable (if ever so slightly). You can see in Figure 1 that our decoder on auto-regressive model severely under-performs beam search, and this is the problem we want to highlight. Using max-rank or even RL trained auto-regressive models, we can observe that we can outperform beam search, and also outperforms the baselines (normalised auto-regressive model). We think that this is a very important result for decoding in text generation, since every time someone wishes to built a decoder that improves over beam search they would always need to solve this issue to actually get improvements on BLEU with their method (results were 1 BLEU point worse before applying these changes). Finally, we also show that this bias is detrimental as models get better, which points out the issues with optimising the decoder for BLEU rather than model score (e.g. https://aclanthology.org/W17-3207.pdf).
> > >
> > > We apologise for the confusing comment, we probably got our ideas mixed up writing the rebuttal. The part about (model score going up and BLEU score going down) refers to model degeneration clearly does not fit here. We would like to point out though, that we do think our approaches address this issue quite competitively compared to some examples you proposed. For instance, https://aclanthology.org/W17-3202.pdf, https://arxiv.org/pdf/2104.05336.pdf use a relatively small search budget (e.g. beam = 50). Also, its important to refer that maxrank doesn't use any more models other than the auto-regressive models, whereas the previously referred work do (e.g. ensembles), which understandably makes the model better, we believe their BLEU score still decreases, which is why there is a cap in the search budget. We show that even with very large search budgets (e.g. MCTS with 256 iterations) we can keep getting BLEU improvements, and beam search seems to not be able to find these translations, which we do think is important finding as well. Of course, we do not expect it to keep improving forever, but it shows that there might be more to decoding in text generation than having a way to calibrate the models to solve some degenerate cases (e.g. search bias), and very deep search (e.g. AlphaGo) might be needed if we are serious about solving the machine translation problem. Unfortunately, BLEU improvements from better search seem to be smaller than modeling improvements, but we do think that if we are serious about solving the machine translation problem, we need to solve the inference problem, otherwise even with perfect models we cannot actually find the correct translation.
> > >
> > > Finally, this one (https://aclanthology.org/W18-6322.pdf) also yields very slow degeneration, we do thank you for pointing it out since we did not know of this.

---

> ### Author Response · Authors · 2021-11-19
> **Rebuttal**
>
> We thank the reviewer for his/her time. You provided a lot of feedback and we do appreciate your effort to make the resulting work stronger. We would like to challenge the reviewer to view this work from a different perspective and clarify some points that are made.
> —
> We agree that we did not cite (https://arxiv.org/pdf/2004.12527.pdf) but this is an Arxiv paper with a questionable experimental section. Their baseline gets 22.32 gets improved to 27.29, but the transformer baseline IWSLT2014 German-English, which are all numbers that are significantly below baseline scores obtained in this dataset (https://paperswithcode.com/sota/machine-translation-on-iwslt2014-german). Thus, we decided to cite https://arxiv.org/pdf/2104.05336.pdf, which is a significantly more elaborate version of this work on a much larger scale. Regarding the novelty of the approach we agree that approaches are similar, but a key difference is that our approach is focused on decoding, while the aforementioned work focuses on training. Thus, aspects like the application of the beam variant of MCTS and the ratio based estimate for uninitialized values (Eq(2)) represent adaptations that we apply to address our specific problem. More importantly, previous work never attempts to address the search bias issue in machine translation (BLEU score decreases as search budget increases), and we provide a solution to this problem (e.g. Max rank). We believe it's unfair to claim that the novelty of our work is limited because other work has used MCTS on language before.
>
> —-
>
> Regarding this comment “The gain brought by the proposed BATS method is overall very small (<=1 BLEU).”, We want to emphasize that the improvements we obtained on a strong baseline, thereby it is expected that the range of improvements are smaller. If you look at Table 1, you can see that the transformer-based auto-regressive model gets 25, and using existing approaches we improved this number to 28.2 using Beam Search. We then use our proposed max Rank feature to further improve this number to 28.6 and then 30 using our decoder. Now, It is significantly harder to improve models that are more optimized because the room for improvement also diminishes. For instance, using our Max Rank feature on a transformer baseline gets an improvement of 1.8 (25.0 to 26.8), but once you train the transformer with MRT the improvement is only 0.4 (28.2 to 28.6). Likewise, when comparing with a weak baseline such as the work in https://arxiv.org/pdf/2004.12527.pdf, you can achieve a 5 BLEU point, which looks significant until you realise the baseline is many times weaker than existing baselines.
>
> Furthermore, we believe that the expected magnitude of the increment in BLEU in modeling work, which covers the majority of work in MT, is not transferable to work on search. In the same way, the expected BLEU improvement of work that crawls parallel data is also different from work in modeling. When crawling parallel data, it is expected that BLEU will increase if enough of it is obtained, so the expectation for a publishable unit is also higher. On the other hand, for decoding in MT, the expectation is that BLEU becomes lower (https://arxiv.org/pdf/2003.03716.pdf) or remains unchanged with better search methods, which is why there are rarely any improvements over beam search. Thus, we find that the fact model scores can be optimized with gains in translation quality is a strong result, which contradicts this prior belief. Furthermore gains are consistent in 3 different languages, which accounts for statistical significance.

---

### Official Review · Reviewer_XcaS · 2021-11-02

**Correctness:** 3
**Technical Novelty And Significance:** 3
**Empirical Novelty And Significance:** 3
**Recommendation:** 6
**Confidence:** 4

**Main Review:**



**Strengths**

1. The motivation is novel and clear.
2. Empirical results verify the efficacy of the method. The findings of the limits of beam search are inspiring. The proposed decoding algorithm could potentially boost the advance of text generation.



**Weaknesses**

1. Despite promising accuracy BATS could achieve, it brings more computational overhead due to rollouts for each node for several iterations. As efficiency is also an important factor to evaluate a translation system, the authors should provide more analyses on this.
2. This experiment part could be further improved in terms of presentation and more clear organization.



**Questions**

1. Page 6, first equation of r. "where we count the number of other actions with lower log-probability than y_i" conflicts with the equation of r where you count the actions that are more probable than y_i. If I am not wrong, you should change "lower" to "larger" in the referred text.

2. The paper claims to generalize among autoregressive and non-autoregressive models. However, I did not find the results of non-autoregressive models or I might be wrong. What do you mean when mentioning a non-autoregressive model? I thought it was that from [Gu+2017] generating a whole sentence in parallel.

**Summary Of The Paper:**

**Summary**

This paper proposes an adaptive tree search algorithm BATS, an MCTS variant, for NMT that could optimize any desired objectives/metrics (e.g. BLEU). The algorithm values each internal node by scoring a greedy searched rollout (instead of that of authentic MCTS) with an autoregressive model, which avoids search biases caused by other heuristics (e.g., beam search would be biased towards shorter translations or incomplete partial generations). Plus, a new objective Max Rank is proposed with a better correlation with translation quality. Experiments show that BART works well in comparison with beam search, which is also shown to bound the progress of more robust modeling in NMT.



**Summary Of The Review:**

This paper inspects the biases of commonly-used decoding heuristics and provides a variant of MCTS algorithm for better decoding in NMT with empirical verification. Overall, I feel like the submission is a good one and would give a concrete contribution to the community. However, the authors should address some of the issues I stated above.

---

> ### Author Response · Authors · 2021-11-19
> **Rebuttal**
>
> We thank the reviewer for his/her time and favorable position towards our paper.
>
> —
> Regarding the computational efficiency of the proposed method, there is a section in the appendix (A.4) that provides such an analysis (you are not forced to read it but we really appreciate it if you did and we can put it into the main paper you find this relevant), where you can see that beam search at 256 is 4 times faster than BATS with 256 iterations. However,  the cost of BATS (~315) with 16 iterations is similar to that of Beam Search with 256 and gets slightly higher model scores.
>
> Furthermore, it shows that the gains BATS gain from increasing the search budget grow steadily, whereas you get diminishing gains as you increase the beam of beam search. This is because BATS is allocating the budget more efficiently. We also explain in section 5.3 that you should only use BATS if you are concerned about maximizing translation quality at a cost of high computational budget. Finally, we think it's a bit unfair that our method is being penalized for being slow. To be clear, beam search is also slow if very large beams are used (e.g. 1024) and even slower if all the 1024 translation hypotheses have to be reranked for the noisy channel models for instance. We consider beam search fast because we generally use it with low beams because there is little incentive to increase the beam size substantially as translation quality plummets, but we show that there are ways to address this issue. Thus, this search problem is an issue that all decoding methods need to address not just BATS. From our analysis, it seems BATS is actually more efficient in this scenario as it obtains better model scores.
>
> —
> Regarding the presentation of the experiments section, could you provide more details or concrete examples on what aspects of the section that could be worked on?
>
> —
> Regarding the first question. You are absolutely right, it should be “lower”, but the < should be replaced with > in the equation. Thanks for noticing.
>
> —
> Regarding the NAR models, we would like to emphasize that a non-autoregressive model is a model where the auto-regressive property (that words are not conditioned on the previously generated words) does not hold. We do understand that there is a lot of work on non-autoregressive models (e.g. https://arxiv.org/abs/1711.02281) where the authors create an instance of a non-autoregressive model by removing the conditional dependency between words in a transformer. However, they are just one instance of what a non-autoregressive model can be. Here, we refer to the noisy channel and Max Rank non-autoregressive models, and all models that use them as components are non-autoregressive as well.
>
> We refer to models guiding the search as non-autoregressive as they require the whole sentence to be completed before it is possible to compute its score and cannot be easily factorized. We generate the sentence by exploring the search tree and use non-autoregressive models to score the greedy continuations.

---

> > ### Comment · Reviewer_XcaS · 2021-11-29
> > **Thanks for your response!**
> >
> > Hi authors, thanks for your detailed response, which basically addressed my comments. After reading other reviewers' comments and author responses, I decided to keep my current evaluation as marginal acceptance.

---

> > > ### Author Response · Authors · 2021-11-29
> > > **Inquiry about the decision**
> > >
> > > Thanks for reading our response. We are a bit surprised about the score we are given as we addressed all your comments.
> > > We noticed that you have written the following in your summary.
> > >
> > > " However, the authors should address some of the issues I stated above.", which we assume is your main reasoning for our current score.
> > >
> > > In our response, we have addressed all issues you have raised. Namely from your weakness section:
> > >
> > > "Despite promising accuracy BATS could achieve, it brings more computational overhead due to rollouts for each node for several iterations. As efficiency is also an important factor to evaluate a translation system, the authors should provide more analyses on this" - We point to the section in the appendix that address this, which also explains that even beam search is slow if we search in such a large search space, while yielding suboptimal results.
> > >
> > > "This experiment part could be further improved in terms of presentation and more clear organization." - Since this is a bit abstract we would like to know what you think is missing in the experimental section.
> > >
> > > Then, in your question section, we have addressed both points. The first one was a typo, which we fixed and deeply appreciate the feedback. In the second case, we explain that the non auto-regressive models are not specific to those used in [Gu+2017] as non-autoregressive models refer to more than just the case used in that line of work. Maybe given this point of view the experimental section may make more sense.
> > >
> > > We do understand that the time of the reviewers is limited, but the purpose of the rebuttal process is allow the authors to address the initial review, so that parts that might not be clear and misunderstood from the reviewers can be made more clear so that the final score more fairly reflects the work that has been submitted. Now given, that the initial position of the reviewer is that the weakness of the paper does not hold, which is what we claim in our rebuttal, and we answer all questions, we think that maintaining the initial score is not fair without further justification.

---

### Official Review · Reviewer_26JG · 2021-11-03

**Correctness:** 3
**Technical Novelty And Significance:** 3
**Empirical Novelty And Significance:** 3
**Recommendation:** 5
**Confidence:** 3

**Main Review:**

The paper studies an important problem, the text generation/decoding algorithm for models. In my view, the paper has two main contributions: utilizing an autoregressive model as the value network to enable MCTS, and proposing a new metric that is not limited to autoregressive decoding. The proposed method achieves better results than beam search as shown in experiments.

I have several concerns:
- The writing of the paper can be largely improved. I find the paper very hard to follow, and some important technical details are missing. For example, about the autoregressive model used as the value network, what is its architecture/training details/datasets/latency when scoring? In equ (3), what is d() and how is it computed? In section 4.1, what are "other actions"? Many notations are not defined, I suggest the authors add a problem definition/notation table at the front of section 3. The proposed method would be easier to understand if an algorithm/(pseudo) code is provided.
- In experiments, what is the exact model and decoding method utilized in the baseline AR/NAR model? Usually NAR models perform worse than AR models, but the conclusion is opposite regarding Table 2. Any explanations? And do you use knowledge distillation to train the NAR model? In NAR models, how the token probability is computed? Still follow the equ in sec 4.1 that uses previous tokens to predict the current one?
- It seems that the proposed method is time-consuming as an AR model is utilized. Have you compared the decoding speed with baseline methods?

Some typos:
- a iterative manner -> an iterative manner
- a set of nodes high value nodes -> a set of high value nodes
- we an observe -> we can observe

This is not a complete list. Seems that the paper is written in a hurry. Please have a carefully check and improve the writing.

=================

I still have the following concerns after reading the rebuttal.

- The incorrect usage of terms and inconsistent claims make the paper hard to follow. I'm still a bit confused regarding the term "non-autoregressive" in the paper. If I understand correctly, in the rebuttal, the authors claim that the "AR/NAR" in Table 2 refers to different objectives such as NC and MR, which means that the backbone models of AR/NAR are the same, and different objectives result in different models. While in the abstract, the authors claim
"Empirically, we show that our adaptive tree search algorithm finds outputs with substantially better model scores compared to beam search in autoregressive models, and compared to reranking techniques in non-autoregressive models."
But NC and MR are not "reranking techniques in non-autoregressive models". Usually used techniques include NPD in Gu et.al., 2018/re-ranking by AT model in Ghazvininejad et.al., 2019. Consider adding comparisons with these techniques?
- The comparison with beam search is unfair. Usually, we set beam=5-6 in practice, which is 20-30 times faster than BATS according to Table 6, while the performance gain is around 1~2%. The price is heavy to achieve marginal gains. The authors claim that they address the issue that the translation quality drops when increasing the beam size in beam search, but in Table 6 BATS also suffers from this issue as the best performance is achieved when iter=2. I'm not sure I understood correctly but I suggest adding further discussions.
- It is surprising that there still exist many typos after "multiple passes over the paper". As pointed above, there exist inconsistent claims over the paper (e.g., the non-autoregressive model or ranking method or objective, the BATS addresses an issue while Table 6 shows opposite results), which makes the paper hard to follow.

Therefore, I'll keep my score and vote for rejection.

**Summary Of The Paper:**

The paper proposes an adaptive tree search algorithm for text generation. It uses an autoregressive model as the value network to produce the playout for each node. While decoding, the paper proposes a metric called max rank to address the shortcomings of utilizing the sum of token level log-probability. Results are conducted on several machine translation datasets, and the proposed method outperforms beam search in most cases.

**Summary Of The Review:**

The paper studies an important problem and proposes a new decoding algorithm based on adaptive tree search, which outperforms beam search in experiments. But the paper writing is poor, some technical details are missed, which hinders the reader from understanding the method. Some metrics are missed in experiments. Therefore I suggest a rejection.

---

> ### Author Response · Authors · 2021-11-19
> **Rebuttal**
>
> We thank the reviewer for his/her review. Unfortunately, we are a bit surprised with the issues raised by the reviewer.
>
> Regarding the dissatisfaction of the reviewer regarding the bad writing quality of the paper, we have done multiple passes over the paper and we were quite satisfied with the way the algorithm is described. It was a bit hard for us to process what the is hard to understand from the examples given:
>
> 1 - In equ (3), what is d() and how is it computed?
> After equation 3, our paper states:
> Where d(p◦y) is initialised as l(p◦y), a function that counts the number of edges required to reach the root node from(p◦y). Then, the following update rule is added to ensure that the d(p) stores that depth of the deepest node achievable from p:
> We define the rule in equation 4 and follow with an explanation of the rule, and an intuitive sense of what d is:
> where we update each node so that d(p) stores the value of the deepest node that is accessible from p. Thus, in Equation 3, the condition d(p◦y)> dmin tests whether p◦y contains a node deeper than dmin.
> Do you mean that the wording is confusing?
> 2 - In section 4.1, what are "other actions"?
> Other actions refer to other words that could have been injected after the prefix. It’s true that “words '' is a more appropriate term so we will change it.
> Finally, we appreciate the suggestion for a notation section, we note that the variables are all introduced in the text, but we will create a notation section to centralise this (We went through the paper again and did not find any missing descriptions when introducing variables and functions).

---

> ### Author Response · Authors · 2021-11-19
> **Rebuttal (cont)**
>
> Regarding the value network model, its parameters and training data are outlined in section 5.1, if this is not clear from the passage, we’re happy to reword it.
>
> > Our autoregressive model transformer baseline uses the multiquery attention model (Shazeer, 2019). It uses the standard architecture with 6 encoder and decoder layers with 512 hidden units, 2048 sized tied embeddings for both source and target word projections and 8 attention heads. We tokenize the data with byte-pair encoding (Sennrich et al., 2016) with 32Kmerges and set a maximum sentence size of Y = 128
>
> We describe in section 2.1 that we use the an autoregressive model
>
> > Therefore, given a node p with prefix y(p) , we computeˆ v(p) by recursively selecting the highest probability token y according to an autoregressive model.

---

> ### Author Response · Authors · 2021-11-19
> **Rebuttal (cont)**
>
> Regarding the NAR models, we would like to emphasize that a non-autoregressive model is a model where the auto-regressive property (that words are not conditioned on the previously generated words) does not hold. We do understand that there is a lot of work on non-autoregressive models (e.g. https://arxiv.org/abs/1711.02281) where the authors create an instance of a non-autoregressive model by removing the conditional dependency between words in a transformer. However, they are just one instance of what a non-autoregressive model can be. Any model that can score a translation (e.g. discriminators, the noisy channel model) is a non-autoregressive model. We explain in the first paragraph in the introduction that auto-regressive component is what allows beam search to be efficient and in the second paragraph we stress the need to challenge the computational issues of removing the auto-regressiveness of the existing models, which is one of the goals of this work. We will try to make it more explicit.
>
> In table 2, we combine different components (auto-regressive model, Max Rank and Noisy Channel model ) using a linear model, using MERT as a way to tune the weights of the linear model, as referred in the experiment section.
> We combine different objective components under a log-linear model, where the weights of the components are tuned with MERT (Och, 2003). In order for model scores to be comparable, all weights are tuned on a pool of 256 translations using beam search.
>
> ​In section 4.1 and 4.2, we explain why the noisy channel model and max rank are non-autoregressive and these are the models we are applying, but we will try to put more emphasis that these do not refer to non-autoregressive models for parallel decoding.

---

> ### Author Response · Authors · 2021-11-19
> **Rebuttal (cont)**
>
> Regarding the computational efficiency of the proposed method, there is a section in the appendix (A.4) that provides such an analysis (you are not forced to read it but we really appreciate it if you did and we can put it into the main paper you find this relevant), where you can see that beam search at 256 is 4 times faster than BATS with 256 iterations. However,  the cost of BATS (~315) with 16 iterations is similar to that of Beam Search with 256 and gets slightly higher model scores.
>
> Furthermore, it shows that the gains BATS gain from increasing the search budget grow steadily, whereas you get diminishing gains as you increase the beam of beam search. This is because BATS is allocating the budget more efficiently. We also explain in section 5.3 that you should only use BATS if you are concerned about maximizing translation quality at a cost of high computational budget. Finally, we think it's a bit unfair that our method is being penalized for being slow. To be clear, beam search is also slow if very large beams are used (e.g. 1024) and even slower if all the 1024 translation hypotheses have to be reranked for the noisy channel models for instance. We consider beam search fast because we generally use it with low beams because there is little incentive to increase the beam size substantially as translation quality plummets, but we show that there are ways to address this issue. Thus, this search problem is an issue that all decoding methods need to address not just BATS. From our analysis, it seems BATS is actually more efficient in this scenario as it obtains better model scores.

---

### Official Review · Reviewer_9nVZ · 2021-11-03

**Correctness:** 4
**Technical Novelty And Significance:** 3
**Empirical Novelty And Significance:** 4
**Recommendation:** 8
**Confidence:** 4

**Main Review:**

Strength:
1. This paper proposes a new beam search algorithm based on  Monte Carlo tree search and compares some methods about modeling improvements to address beam search bias.

2. The baselines are strong and the proposed approach also yields BLEU improvements when the model score correlates well with BLEU.

Minor weakness:
1. The paper does not intensively analyze the running efficiency of the proposed BATS algorithm. Additionally, it would be better to discuss some ways to improve the its efficiency, which is helpful to make it practical in future.
2. In the paper, max rank is claimed to be a contribution but it does not conduct an ablation study to highlight its individual contribution. For example, in Table 1, it would be better to add one row by setting MR=Yes whereas NC and MRT are No.
3. It shows the BLEU comparison for standard beam search and BATS, but it does not show the comparison in terms of model score. For example, in the line 1 of Table 1, does BATS obtains better model score than Beam search even if its BLEU is worse?

Typo:
1. we an observe a large gap between =》 we can observe ...



**Summary Of The Paper:**

This paper proposes an adaptive tree search algorithm for NMT models. One advantage of this algorithm is that it does not make any assumptions about search objectives, and this enables the proposed algorithm to be applied on top of more general search objectives. In addition, it studies the issue of beam search bias and revisits some tricks to alleviate it as well as a new tricks. Combined with these tricks, the proposed algorithm delivers clear BLEU improvements over a strong baseline.


**Summary Of The Review:**

This paper proposes an adaptive tree search algorithm for NMT models, and the proposed algorithm yields BLEU improvements when the model score correlates well with BLEU.

---

> ### Author Response · Authors · 2021-11-19
> **Rebuttal**
>
> We thank the reviewer for their time and favorable position towards our paper.
>
> Regarding the summary, we would like to point out that the search bias is latent to beam search, and MCTS does not suffer from it and but different biases. The problem we are solving with our approach is a modeling problem, which favors the search bias as degenerate solutions are eliminated. We present this in different ways throughout the paper but we make this clear in the passage in section 4:
> “As decoding with a decoder on vanilla autoregressive models is unlikely to yield translations with
> quality superior to beam search, as the beam search bias is essential to overcoming the calibration
> issues in these models, we propose modeling improvements in order to address these shortcomings.”
>
> Please let us know if anything in the paper is unclear in this passage, or if you were asking a different question.

---

> ### Author Response · Authors · 2021-11-19
> **Rubuttal (cont)**
>
>
> Regarding the ablation test with MaxRank, we provide this result in Table 2 for Chinese-English, with the comment in section 5.3:
>
> In the models where beam search outperforms BATS(Rows “Log ProbabilityAR(α= 0)” and “Log ProbabilityAR(α= 0.8)”), we notice that the search budget in both cases is always low.
>
> Then, in the next paragraph we describe what happens when we add the MaxRank feature:
> With the addition of the noisy channel model (Row “Log Probability (α=0.8) + NCNAR) and Max Rank (Row “Log Probability (α = 0.8) + MRNAR”), the search budget until BLEU deteriorates for both decoders is increased. Here, we observe that BATS is the decoding option that yields higher BLEU and that the delta is higher when BATS can employ more iterations.
>
> This is the case scenario you describe, and all combinations of different tag flips are in Table 2. We restricted this to Chinese-English to not overload the paper with information, but if you prefer we can add all results in the appendix.

---

> ### Author Response · Authors · 2021-11-19
> **Rebuttal (cont)**
>
>
> Regarding the comparison between Beam Search and BATS in terms of model score, results are illustrated in Figure 1. We also describe the models scores in the following passage in section 5.3:
>
> We observe in Figure 1 that the evolution of BLEU is considerably more stable with the noisy channel
> model (Column “Log Probability (α= 0.8) + NC”) and when Max Rank is applied (Column “Log Probability (α = 0.8) + MR”). For these non-autoregressive models BATS yields significantly better model scores than reranking. More interestingly, comparing the behavior of Beam Search and BATS, similar model scores between the two methods do not yield similar BLEU scores.  In the noisy channel model, we observe that at 16 iterations and beam size 16, both decoders yield similar model scores and BLEU scores. However, as the model score increases beyond that point, the degeneration in beam search as the score increases is significant, while BATS observes almost no deterioration. This suggests that the beam search bias has a negative impact in translation quality at high values  by filtering good translations from the search space. Using Max Rank, we observe that not only BATS can achieve considerably higher model scores as it is optimizing the non-autoregressive model directly, it also yields considerably higher BLEU scores. In beam search, BLEU stops improving at 32 iterations even though the model score keeps increasing due to the search bias.
>
> So not only beam search yields lower scores but they lead to worse translation quality when the scores are similar, which is the result of the search bias negatively impacting the translation quality.
> We could add scores in table 2 but we think they would be a bit misleading since they are not optimal and could be increased if the translation budget is higher.

---

> ### Author Response · Authors · 2021-11-19
> **Rebuttal (cont)**
>
> Regarding the computational efficiency of the proposed method, there is a section in the appendix (A.4) that provides such an analysis (you are not required to read it but it will clarify things, and we can put it into the main paper you find this relevant), which shows that beam search at 256 is 4 times faster than BATS with 256 iterations. However, the cost of BATS (~315) with 16 iterations is similar to that of Beam Search with 256 and already gets higher model scores, which provides another favorable point of comparison.
>
> Furthermore, it shows that the gains BATS gain from increasing the search budget grow steadily, whereas you get diminishing gains (and even reversals!) as you increase the beam of beam search. This is because BATS is allocating the budget more efficiently and is less biased. We also explain in section 5.3 that you should only use BATS if you are concerned about maximizing translation quality at a cost of high computational budget. Finally, we think it's a bit unfair that our method is being penalized for being slow. To be clear, beam search is also slow if very large beams are used (e.g. 1024) and even slower if all the 1024 translation hypotheses have to be reranked for the noisy channel models for instance. We consider beam search fast because we generally use it with low beams because there is little incentive to increase the beam size substantially as translation quality plummets, but we show that there are ways to address this issue. Thus, this search problem is an issue that all decoding methods need to address not just BATS. From our analysis, it seems BATS is actually more efficient in this scenario as it obtains better model scores.

---

### Decision · Program_Chairs · 2022-01-20

**Decision:**

Accept (Poster)

**Comment:**

This paper proposes an adaptive tree search algorithm for NMT models with non-decomposable metrics and shows its efficacy against strong baselines. This is an interesting contribution towards overcoming the performance caps introduced by the uncontrolled-for biases of beam search, and it speaks to a growing community interested in decoding beyond greedy surprisal minimisation.

The initial reviews brought to light a number of concerns that in my view are well addressed in the rebuttal and in the current version of the manuscript. One of the key issues was a confusion caused by the use of the term 'non-autoregressive' to refer to the intractability of the metric / objective function of certain models. This use clashed with the more standard use in MT, which refers to a tractable factorisation of a joint probability by means of strong conditional independence assumptions.

The confusion is easy to address and in no way compromises the thoroughness of the empirical section. The authors are aware of the confusion and how to resolve it, and they have acknowledged the need to pick a less ambiguous term.

I'd like to recommend this for acceptance, but I urge that the authors do not ignore the confusion caused by 'auto/non-auto regressive' and the missing literature that came up in the discussion with reviewer i2pz (I understand the discussion happened too late for the manuscript to be updated, but I trust this can be done for the final version).